# Improving Machine Learning Diabetes Prediction Models for the Utmost Clinical Effectiveness

**DOI:** 10.3390/jpm12111899

**Published:** 2022-11-14

**Authors:** Juyoung Shin, Joonyub Lee, Taehoon Ko, Kanghyuck Lee, Yera Choi, Hun-Sung Kim

**Affiliations:** 1Health Promotion Center, Seoul St. Mary’s Hospital, Seoul 06591, Korea; 2Division of Endocrinology and Metabolism, Department of Internal Medicine, Seoul St. Mary’s Hospital, College of Medicine, The Catholic University of Korea, Seoul 06591, Korea; 3Department of Medical Informatics, College of Medicine, The Catholic University of Korea, Seoul 06591, Korea; 4Department of Biomedicine and Health Sciences, College of Medicine, The Catholic University of Korea, Seoul 06591, Korea; 5NAVER CLOVA AI Lab, Seongnam 13561, Korea

**Keywords:** diabetes prediction model, diabetes prevention, type 2 diabetes, XGBoost Survival Embedding

## Abstract

The early prediction of diabetes can facilitate interventions to prevent or delay it. This study proposes a diabetes prediction model based on machine learning (ML) to encourage individuals at risk of diabetes to employ healthy interventions. A total of 38,379 subjects were included. We trained the model on 80% of the subjects and verified its predictive performance on the remaining 20%. Furthermore, the performances of several algorithms were compared, including logistic regression, decision tree, random forest, eXtreme Gradient Boosting (XGBoost), Cox regression, and XGBoost Survival Embedding (XGBSE). The area under the receiver operating characteristic curve (AUROC) of the XGBoost model was the largest, followed by those of the decision tree, logistic regression, and random forest models. For the survival analysis, XGBSE yielded an AUROC exceeding 0.9 for the 2- to 9-year predictions and a C-index of 0.934, while the Cox regression achieved a C-index of 0.921. After lowering the threshold from 0.5 to 0.25, the sensitivity increased from 0.011 to 0.236 for the 2-year prediction model and from 0.607 to 0.994 for the 9-year prediction model, while the specificity showed negligible changes. We developed a high-performance diabetes prediction model that applied the XGBSE algorithm with threshold adjustment. We plan to use this prediction model in real clinical practice for diabetes prevention after simplifying and validating it externally.

## 1. Introduction

In Korea, general medical checkups are popular owing to the mandate of the government and the financial aid provided by companies. These medical checkups consist of various examinations, including laboratory, imaging, and functional studies, followed by individualized recommendations [1,2,3]. Previously, people visited hospitals for diagnosis and treatment after symptom occurrence. However, nowadays, many people visit hospitals for screening even without symptoms in the hopes of early disease identification followed by timely treatment and a good prognosis [4]. Furthermore, people desire a reliable prediction of their conditions and short- and long-term possibilities of disease occurrence followed by appropriate intervention to prevent or delay them. They consider that recent technological advances have led to the development of new methods that can provide better answers. Diseases occur from a combination of genetic and environmental/external factors [5]. Many chronic diseases, such as diabetes mellitus (DM), hypertension, and obesity, are closely related to patient lifestyle choices, including nutrition and physical activity, and their occurrence cannot be attributed to a single factor.

DM is a major chronic disease with a growing prevalence [6]. The management of diabetes and its related complications consumes valuable social resources. Therefore, strategies to prevent diabetes are as important as an early diagnosis, and the development of a diabetes predictive model can be the first step toward performing the necessary interventions [7]. By informing each individual of their risk for diabetes, healthcare professionals can prescribe the appropriate individualized lifestyle modifications and monitor their progress. 

Machine learning (ML) algorithms have been utilized for developing predictive models in various fields, such as weather forecasting, predicting consumer propensity, and traffic prediction [8,9,10,11,12]. Furthermore, ML has been applied in the medical field; for instance, it is used to explain disease progression to patients. Many models have been created to predict the possibility of diseases, and they are expected to suggest appropriate interventions after stratifying the risk [13,14]. In a previous study, we made simplified ML models for 12-month DM predictions [15]. In this study, we aimed to develop more valid and applicable DM prediction models for longer periods by comparing the performances of widely used ML models, such as classification and decision tree-based algorithms. To this end, we applied logistic regression for the former and decision tree, random forest, and eXtreme Gradient Boosting (XGBoost) models for the latter. XGBoost is a decision-tree-based ensemble ML algorithm whose good performance has been widely demonstrated in ML competitions [16,17]. Subsequently, we compared the recently introduced open-source algorithm based on XGBoost, the XGBoost Survival Embedding (XGBSE) model [18], against the conventional Cox regression model.

## 2. Materials and Methods

### 2.1. Privacy Protection

All the data extracted from electronic medical records for this study were separately stored in secure computing systems and facilities, and access was provided solely to the principal investigator. Examinees were assigned random temporary identifications, and only encrypted, anonymous data were available to observers and analysts. Because the data used in this study were anonymized, no violation of human rights or infringement of moral or ethical rights was possible. Consequently, it was not necessary to obtain informed consent from the participants. The study was approved by the Institutional Review Board of the Hospital (IRB No. KC20RISI900).

### 2.2. Data Collection

The dataset used in this research was extracted from electronic medical records between July 2009 and April 2019 at the Health Promotion Center of a tertiary hospital in Seoul, Korea. The general checkups were open to all without any restrictions, such as occupation or underlying diseases. Regular checkups were recommended, generally in the form of annual follow-ups; however, the interval depended on the examinees’ intentions. Data from only the adult subjects (18 years of age or older) were collected. First, we selected subjects with at least two medical checkups at intervals of three or more months. Then, we excluded those who were diagnosed with diabetes at the first visit. We defined patients with diabetes if they fulfilled any of the following three criteria [19]: (1) self-reported diabetes, (2) taking any glucose-lowering agent, and (3) fasting glucose level ≥ 126 mg/dL and glycosylated hemoglobin (HbA1c)  ≥  6.5%.

The medical records contained information on approximately 99,414 individuals with 206,398 checkups (Figure 1). A total of 59,194 individuals with only one visit and 1841 individuals who had diabetes at their first visit were excluded from the study, and the remaining 38,379 individuals were included. Among them, 1518 patients were diagnosed with diabetes, whereas 36,861 were not. After random splitting, the training dataset consisted of 29,489 subjects with no diabetes and 1214 subjects with incident diabetes.

### 2.3. Variables Used to Develop the Diabetes Prediction Model

The variables included in this study are presented in Appendix A. These included age, sex, underlying diseases, family history, physical examinations, and laboratory results. Additionally, systolic blood pressure (SBP) and diastolic blood pressure (DBP) were included. The anthropometric variables of ideal body weight, skeletal muscle mass, body fat mass, and percent body fat were measured through a bioelectrical impedance analysis performed using the InBody 720 device (Biospace, Seoul, Korea). The body mass index (BMI) was calculated by dividing the patient’s weight (kg) by the height squared (m^2^). The data of the pulmonary function test, including forced vital capacity (FVC), forced expiratory volume in 1 s (FEV1), forced expiratory flow at 25% and 75% of vital capacity (FEF25–75), and peak expiratory flow rate (PEFR), were available. Among the laboratory tests, we used the data points under the following variables: fasting glucose (FBG), glycosylated hemoglobin (HbA1c), total cholesterol, triglyceride, high-density lipoprotein cholesterol (HDL-C), low-density lipoprotein cholesterol (LDL-C), hemoglobin, hematocrit, total protein, albumin, uric acid, aspartate transaminase, alanine aminotransferase, alkaline phosphatase, amylase, gamma-glutamyl transpeptidase, globulin, albumin/globulin ratio, blood urea nitrogen, creatinine, sodium, potassium, chloride, calcium, phosphorus, thyroid stimulating hormone, and free thyroxine. As the questionnaires asked comprehensive questions about underlying diseases and family history, the total number of variables was 112, and the variables comprised numerical and categorical values.

### 2.4. Construction and Validation of the DM Prediction Models

We split the dataset into training and testing datasets by randomly assigning 80% and 20% of the data points to each dataset, respectively. Applying the stratified 10-fold cross-validation method to the training dataset, we tuned the model hyperparameters to obtain optimal hyperparameters that would yield the largest area under the receiver operating characteristic (AUROC). Missing values were replaced with the median and mode values in the case of the continuous and discrete features, respectively. Algorithms that could handle missing values produced two different predictive models with and without the imputation of missing values using the medians and modes. The values of Shapley additive explanations (SHAP) were used to determine the importance of each variable. First, the logistic regression, decision tree, random forest, and XGBoost models were applied to predict the presence of diabetes within 9 years using a checkup record at the first visit for each person. Then, the Cox regression and XGBSE models were applied to calculate the possibility of diabetes within a certain period using all serial data for survival analysis. We used the AUROC, sensitivity, specificity, positive predictive value (PPV), negative predictive value (NPV), and accuracy to evaluate the performance of each model. The AUROC is a performance metric between sensitivity and specificity. Sensitivity and specificity are the abilities used to correctly predict whether an individual has diabetes or does not have diabetes, respectively. PPV and NPV are the percentages of true incident diabetes and no diabetes, respectively. 

## 3. Results

The characteristics of the study population are summarized in Table 1, and those used in the test data are presented in Appendix A. The mean age of the subjects of the training set was 45 ± 10 years, and 44.2% of the subjects (13,578/30,703) were female. The mean BMI was 23.4 ± 3.2 kg/m^2^. Laboratory test results showed that the mean FBG was 91 ± 12 mg/dL, and HbA1c was 5.4 ± 0.3%. The mean duration of the follow-up was 3.9 years, and the mean number of medical checkups was 3.70. Numerous differing characteristics were observed between the incident diabetes and “no diabetes” groups. Subjects in the incident diabetes group were older (51 ± 10 years vs. 45 ± 10 years), and a lower proportion of females was observed therein (26.6% vs. 44.9%). Anthropometric measurements showed that people in the incident diabetes group had higher BMI (25.8 ± 3.4 kg/m^2^ vs. 23.3 ± 3.1 kg/m^2^), body fat percentage (27.8 ± 6.3% vs. 26.0 ± 6.1%) and waist–hip ratio (0.93 ± 0.08 vs. 0.88 ± 0.12) than those in the “no diabetes” group. Either SBP (127 ± 14 mmHg vs. 118 ± 14 mmHg) and DBP (78 ± 11 mmHg vs. 73 ± 10 mmHg) or FBG (109 ± 21 mg/dL vs. 90 ± 10 mg/dL) and HbA1c (6.0 ± 0.3 vs. 5.4 ± 0.3%) were higher in the incident diabetes group. The average number of visits for the incident diabetes and “no diabetes” groups was 4.66 and 3.67, respectively, and the follow-up periods were 5.4 years and 3.9 years, respectively. Overall, the incidence rate of diabetes was 3.96%.

Figure 2 shows the effects of each variable on diabetes prediction and lists the top 20 variables in decreasing order of importance. FBG and HbA1c achieved remarkable importance among all the variables, followed by the family history of diabetes, age, waist–hip ratio, triglyceride, TSH, HDL-C, FVC, etc. The mean SHAP values of FBG and HbA1c were four and three times those of the third-ranked variable, family history of diabetes (0.963 for FBG, 0.761 for HbA1c, and 0.244 for the family history of diabetes). All four measurements of the pulmonary function test, FVC, PEFR, FEF 2575%, and FEV1, were included among the top 20 important variables.

Table 2 shows the performance parameters of several different prediction models that determined the existence of diabetes for the 9-year period. All seven models showed high accuracies; all accuracy values exceeded 0.9. The sensitivities were high, but the specificities were low as a result of the trade-off relationship between sensitivity and specificity. The AUROC ranged from a minimum of 0.524 in the random forest model with a missing value imputation by median to a maximum of 0.623 in the XGBoost model without missing value imputation. 

Table 3 shows the performance parameters of the XGBSE models, which determined the existence of diabetes in each corresponding period from 2 years to 9 years. The XGBSE models yielded remarkably high AUROCs; each model achieved an AUROC exceeding 0.9. Among them, the 9-year prediction XGBSE model had the highest AUROC of 0.955. The sensitivity of our XGBSE models for the standard threshold value of 0.5 ranged between 0.090 and 0.818, whereas the specificity ranged between 0.917 and 0.999. The models for longer-term predictions achieved higher sensitivity values. We lowered the threshold to 0.25 to increase sensitivity; subsequently, the sensitivity changed from 0.090 to 0.326 and from 0.818 to 0.934 for the 2- and 9-year prediction models, respectively. The specificity range decreased from between 0.917 and 0.999 to between 0.792 and 0.992. When the XGBSE model was compared with the Cox regression model, C-indexes of 0.934 and 0.921 were obtained for the XGBSE and the Cox regression model, respectively. 

## 4. Discussion

In this study, using the recently developed XGBSE algorithm, we created a model to predict diabetes development from electronic medical records data, which were created based on general medical checkups. Each XGBSE model performed predictions over various periods ranging from 2- to 9-years, and the models outperformed other existing models. Considering the results of a recent systematic review and meta-analysis [20] that suggested an AUROC of 0.812 for ML prediction models, our prediction models with AUCs exceeding 0.9 are satisfactory. 

One potential reason for the excellent performance of our model is that we used as many variables as possible, including self-reported information, physical examinations, and laboratory tests, while excluding complicated information, such as the results of imaging studies or unavailable data. Although the use of more variables leads to a complex and cumbersome process, this process facilitates a high predictive power for the model [21]. Other meaningful variables, such as medication history, smoking, alcohol consumption, stress, occupation, and education, were unavailable [22,23,24,25]. However, we included the two most important variables of FBG and HbA1c levels. Previous studies have shown the importance of these two parameters for diabetes prediction models [15,26], and the estimation of variable importance in this study confirmed the high importance of FBG and HbA1c levels. Although the order of importance was different, the family history of diabetes, age, TG, BMI, and uric acid were also meaningful variables, and the results were consistent with those of other studies [27,28,29]. Our model included distinctive variables such as waist–hip ratio and the results of pulmonary function tests, which are not used in other deep learning models and were considered contributing factors for outstanding performance. 

In our study, the waist–hip ratio had higher SHAP values than those of BMI. Although the choice of the best anthropometric index remains debatable [30,31], these are well-known factors relating to diabetes [32,33,34]. Shin et al. developed an ML-based diabetes prediction model using the results of pulmonary function tests [15]. Other studies have reported a close relationship between pulmonary function and diabetes incidence [35,36,37,38], and our study showed that FVC, PEFR, FEF 25–75%, and FEV1 were included in the top 20 important variables. Here, we determined the importance of all the variables with SHAP values, unlike the previous study by Shin et al., in which feature importance proved the effect of 27 selected variables to fit the simplified model [15]. In addition to the advantages offered by a large number of variables used in our study, our data source and the records of regular medical checkups offered the advantage of a small number of missing values. The examinees were active subjects of these checkups; hence, they sincerely answered the questionnaires and participated in the tests. In fact, based on the SHAP values, the third most important variable in this study was a family history of diabetes. A third critical factor accounting for the success of our model could be the use of a threshold of 0.25, which increased the sensitivity with negligible changes in specificity, even though sensitivity and specificity have a trade-off relationship. It has been noted that an optimal threshold choice is essential to improve the performance of artificial intelligence algorithms [39]. This manipulation of higher sensitivity is reasonable as the fundamental goal of this diabetes prediction model was to reduce the economic burden by preventing diabetes rather than initiating medication. There is negligible harm in diagnosing the risk of developing diabetes and recommending lifestyle modifications to prevent it. 

The final important point accounting for the excellent performance of our model might be the XGBSE algorithm. Chen et al. [16] proposed a decision-tree-based XGBoost as an effective and flexible ML method. This model has also been introduced in the clinical field [40,41,42,43]. XGBSE is an open-source, and a state-of-the-art modeling algorithm developed to solve concerns about survival-curve prediction, confidence-interval estimation, and unbiased expected survival times [18]. To the best of our knowledge, this is the first study that uses the XGBSE algorithm for diabetes prediction. The performance of our model with the XGBSE algorithm showed a higher AUROC than that of XGBoost. Generally, achieving accurate predictions for longer prediction periods is difficult. Rhee et al. compared the Cox longitudinal summary and deep learning models for up to 10 years and obtained a declining AUROC value [44]. Our models maintained good AUROC values across the 2- to 9-year periods; this is another notable strength as our models predict serially for prediction survival. 

The ultimate goal of these diabetes prediction models is to identify the people at risk for diabetes and motivate them to incorporate changes into their lifestyles for better health. In Korea, people voluntarily undergo regular comprehensive medical checkups. Owing to technological advancements, people’s expectations are shifting from early diagnosis and treatment to an early detection of disease possibility and preventive management. To address these expectations, commercialized genetic studies were introduced. However, not only genetics but also environmental and/or external conditions can influence disease progression, and currently available commercialized genetic studies are unable to reflect changes in people’s conditions. In addition, everyone undergoes internal aging, and the risk of diabetes increases with age. Furthermore, our data show that age is an important variable in predicting diabetes (Figure 2). These factors appear to account for the differences—not only age but also the number of checks or lengths of follow-up periods—between the incident diabetes group and the no diabetes group (Table 1). Therefore, disease prediction models will be helpful to guide people, particularly those with chronic diseases, such as diabetes. Awareness concerning the risk of diabetes can motivate people to track changes in their risk after applying preventive measures. In the near future, the results of the ongoing Korean Diabetes Prevention Study can serve as an evidence-based guide for determining suitable interventions to prevent diabetes [45]. In addition, healthcare providers and politicians can formulate strategies for a stratified approach to retard the progress of DM. These efforts can be implemented not only by healthcare providers but also target high-risk individuals through various types of support, including official educational programs, smartphone applications, and social networks. 

This study has some limitations. Our dataset was obtained from a tertiary institution. Typically, patients with severe and multiple diseases tend to visit higher-level hospitals. However, this tendency is expected to be lower because the data were obtained from a disease screening service where examinees volunteered for general checkups. Secondly, the models were only validated internally. We cannot confirm that our study subjects are representative of the general population [46,47]. Therefore, we plan to perform external validations before applying the model in other scenarios. Excessive numbers of parameters and variables to evaluate can also be a limitation.

## 5. Conclusions

We developed a high-performance model to predict diabetes via various prediction periods ranging from 2- to 9-year periods based on a large, reliable dataset that included numerous variables, such as answers to a questionnaire and the results of physical, functional, and laboratory examinations after comparing various ML algorithms via threshold adjustment. We hope this diabetes risk predictor will set a foundation for diabetes prevention with appropriate individualized and specified interventions.

## Figures and Tables

**Figure 1 jpm-12-01899-f001:**
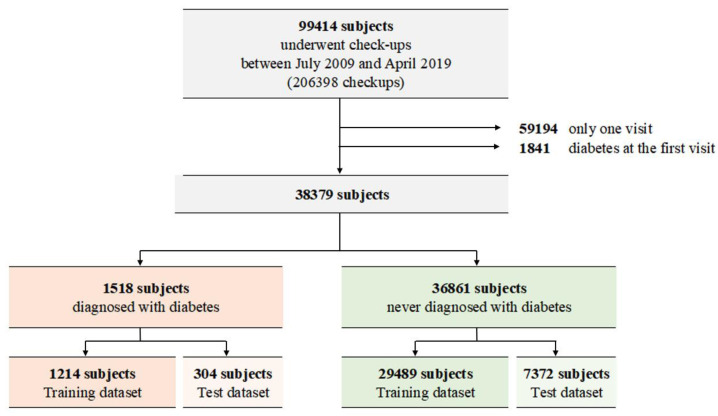
Description of the enrolled subjects.

**Figure 2 jpm-12-01899-f002:**
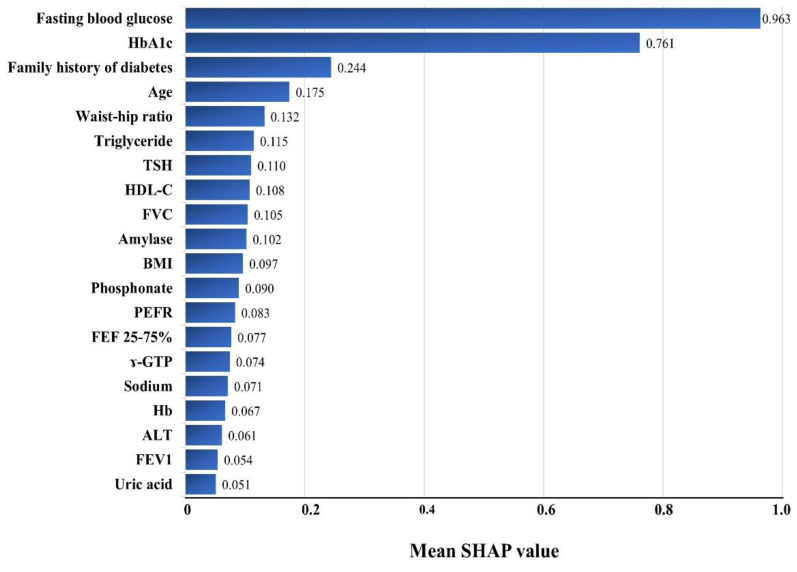
Variable importance in the XGBSE diabetes risk prediction models using the Shapley additive explanations (SHAP) approach. HbA1c—glycosylated hemoglobin; TSH—thyroid stimulating hormone; HDL—high-density lipoprotein; FVC—forced vital capacity; BMI—body mass index; PEFR—peak expiratory flow rate; FEF—forced expiratory flow; GTP—glutamyl transpeptidase; Hb—hemoglobin; ALT—alanine aminotransferase; FEV1—forced expiratory volume in 1 s; SHAP—Shapley additive explanations.

**Table 1 jpm-12-01899-t001:** Characteristics of the subjects in training data.

	All (*n* = 30,703)	Incident Diabetes (*n* = 1,214)	No Diabetes (*n* = 29,489)
Age, years	45 ± 10	51 ± 10	45 ± 10
Sex, female (%)	13,578 (44.2)	323 (26.6)	13,255 (44.9)
BMI, kg/m^2^	23.4 ± 3.2	25.8 ± 3.4	23.3 ± 3.1
Body fat percent, %	26.1 ± 6.2	27.8 ± 6.3	26.0 ± 6.1
Waist-hip ratio	0.88 ± 0.12	0.93 ± 0.08	0.88 ± 0.12
SBP, mmHg	119 ± 14	127 ± 14	118 ± 14
DBP, mmHg	73 ± 10	78 ± 11	73 ± 10
Laboratory finding			
FBG, mg/dL	91 ± 12	109 ± 21	90 ± 10
HbA1c, %	5.4 ± 0.3	6.0 ± 0.3	5.4 ± 0.3
Total cholesterol, mg/dL	195 ± 34	200 ± 38	195 ± 33
Triglyceride, mg/dL	112 ± 80	162 ± 109	110 ± 78
HDL-C, mg/dL	54 ± 13	47 ± 11	54 ± 13
LDL-C, mg/dL	118 ± 30	122 ± 34	118 ± 30
AST, IU/L	23 ± 13	28 ± 15	23 ± 13
ALT, IU/L	25 ± 21	36 ± 27	25 ± 21
gamma-GTP, IU/L	34 ± 39	54 ± 50	34 ± 38
Amylase, IU/L	86 ± 29	79 ± 30	87 ± 29
BUN, mg/dL	13 ± 4	14 ± 4	13 ± 4
Creatinine, mg/dL	0.9 ± 0.2	0.9 ± 0.4	0.9 ± 0.2
Sodium, mEq/L	142 ± 2	142 ± 2	142 ± 2
Potassium, mEq/L	4.2 ± 0.3	4.2 ± 0.3	4.2 ± 0.3
Calcium, mg/dL	9.1 ± 0.4	9.2 ± 0.4	9.1 ± 0.4
Phosphate, mg/dL	3.5 ± 0.5	3.5 ± 0.5	3.5 ± 0.5
Uric acid, mg/dL	5.4 ± 1.4	5.9 ± 1.4	5.4 ± 1.4
TSH, uIU/ml	2.24 ± 4.97	2.25 ± 3.42	2.24 ± 5.03
Hemoglobin, g/dL	14.4 ± 1.6	14.9 ± 1.5	14.7 ± 1.6
Pulmonary function test			
FVC, %	93.8 ± 11.0	90.9 ± 11.1	92.9 ± 10.9
FEV1, %	98.6 ± 12.8	97.7 ± 13.3	98.7 ± 12.8
FEV1/FVC, %	82.4 ± 6.7	80.7 ± 6.1	82.4 ± 6.8
FEF25–75, %	100.1 ± 26.9	97.9 ± 27.8	100.2 ± 26.8
PEFR, L/s	99.1 ± 17.5	99.5 ± 18.3	99.1 ± 17.5
Personal history			
Hypertension (%)	3320 (11.0)	319 (26.5)	3001 (10.3)
Cardiovascular diseases (%)	449 (1.5)	55 (4.6)	394 (1.4)
Cerebrovascular diseases (%)	371 (1.2)	20 (1.7)	351 (1.2)
Family history			
Hypertension (%)	12,797 (42.1)	577 (47.8)	12,220 (41.9)
Diabetes (%)	8927 (29.4)	604 (50.0)	8323 (28.5)
Cardiovascular diseases (%)	5529 (18.2)	252 (20.9)	5277 (18.1)
Cerebrovascular diseases (%)	6907 (22.7)	338 (28.0)	6569 (22.5)
Follow-up, year	3.9 ± 2.5	5.4 ± 2.4	3.9 ± 2.5
Checkup, *n*	3.7 ± 2.0	4.7 ± 2.3	3.7 ± 2.0

Categorical variables are reported as frequencies (%), and continuous variables are reported as mean ± SD. BMI—body mass index; SBP—systolic blood pressure; DBP—diastolic blood pressure; FBG—fasting blood glucose; HbA1c—glycosylated hemoglobin; HDL-C—high-density lipoprotein cholesterol; LDL-C—low-density lipoprotein cholesterol; AST—aspartate transaminase; ALT—alanine aminotransferase; gamma-GTP—glutamyl transpeptidase; BUN—blood urea nitrogen; TSH—thyroid stimulating hormone; FVC—forced vital capacity; FEV1—forced expiratory volume in 1 s; FEF25–75—forced expiratory flow at 25% and 75% of vital capacity; PEFR—peak expiratory flow rate.

**Table 2 jpm-12-01899-t002:** Performance parameters of diabetes prediction models.

		AUROC	Sensitivity	Specificity	PPV	NPV	Accuracy
**Logistic regression**	**MI by Median**	0.547	0.964	0.529	0.996	0.098	0.960
**Decision tree**	**MI by Median**	0.590	0.967	0.582	0.994	0.186	0.962
	**No MI**	0.598	0.968	0.556	0.993	0.204	0.962
**Random Forest**	**MI by Median**	0.524	0.962	0.696	0.999	0.050	0.961
	**No MI**	0.530	0.962	0.756	0.999	0.061	0.962
**XGBoost**	**MI by Median**	0.616	0.969	0.646	0.994	0.237	0.964
	**No MI**	0.623	0.970	0.690	0.995	0.250	0.966

AUROC—area under the receiver operating characteristic curve; NPV—negative prediction value; PPV—positive prediction value; MI—missing value imputation.

**Table 3 jpm-12-01899-t003:** Diabetes risk prediction model performance evaluation parameters by XGBoost Survival Embedding method.

		Threshold = 0.5						Threshold = 0.25				
Prediction Period	AUROC	Sensitivity	Specificity	PPV	NPV	Accuracy		Sensitivity	Specificity	PPV	NPV	Accuracy
2 year	0.949	0.090	0.999	0.571	0.985	0.984		0.326	0.992	0.387	0.989	0.981
3 year	0.953	0.190	0.996	0.600	0.974	0.970		0.465	0.984	0.489	0.982	0.967
4 year	0.945	0.332	0.989	0.643	0.962	0.953		0.626	0.969	0.536	0.978	0.95
5 year	0.948	0.377	0.985	0.692	0.947	0.936		0.686	0.960	0.599	0.972	0.938
6 year	0.945	0.506	0.982	0.804	0.932	0.922		0.741	0.948	0.674	0.962	0.922
7 year	0.937	0.578	0.974	0.846	0.905	0.897		0.785	0.926	0.720	0.947	0.898
8 year	0.940	0.671	0.97	0.925	0.841	0.863		0.856	0.888	0.809	0.917	0.876
9 year	0.955	0.818	0.917	0.969	0.615	0.842		0.934	0.792	0.934	0.792	0.900

AUROC—area under the receiver operating characteristic curve; NPV—negative prediction value; PPV—positive prediction value.

## Data Availability

The data that support the findings of this study are available from the corresponding author upon request.

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
