# Peer review of "Improving Machine Learning Diabetes Prediction Models for the Utmost Clinical Effectiveness"

_jpm, 2022, doi:10.3390/jpm12111899_

Round 1

Reviewer 1 Report

In this draft, Shin and others evaluate several machine learning models in predicting diabetes mellitus (DM) using data collected at a regional hospital in Seoul, South Korea. The dataset used in this study consists medical records from over 30,000 patients who had at least two regular check-ups and were not clinically diabetic at their first visits. These health records seem to be well documented and cover a comprehensive set of variables, some of which are known predictors of DM (fasting blood glucose, HbA1c, waist size etc.). The authors have evaluated both regression models and survival analysis models, for the latter they tested both the conventional proportional hazards model and a recently emerged model, XGBSE, that has gained popularity in the field of long-term disease prediction, though this method is probably too new to be fully validated for real-world applications. The authors concluded that XGBoost, either used as a straightforward “single-measurement” predictor or used to enable survival analysis with XGBSE, outperformed other models in DM prediction on their dataset.

Overall, the study was well executed, and the dataset was quite impressive. It is a good validation case for XGBSE, which could be found interesting by readers of the Journal of Personalized Medicine. That being said, I do have a few major concerns about the draft manuscript:

1.     Even though the train-test split is random, given the relatively low numbers of diabetic patients enrolled in the study, it is important to show that the training and the test datasets are comparable in dataspace. The authors could consider appending the side-by-side feature comparisons of the training and test dataset as they did for Table 1. 

2.     The authors did not specify how exactly was the “single-measurement” training performed. Each patient should have multiple entries and for the uncensored ones who developed DM within the data collection period, the intervals between time-to-first-visit and time-to-DM-diagnosis may vary greatly. Did the authors use the patient’s data of the first visit or the aggregated data of all visits to predict the development of DM over the 9-year period? Or each entry was treated differently, and the prediction was made with the paired check-up data and the DM diagnoses? If it’s the latter, I would hard call it a “predictive” model as it is more like a “diagnosis” model. Although I could have misunderstood the terminologies here and it would be greatly appreciated if the authors can clarify. 

3.     I don’t quite understand the term threshold used to tune the sensitivity of the XGBSE model: do the authors mean by the estimated probability of survival without DM by year X? If so, please specify in the main text as well as in the method section.

Other comments:

4.     In table 2, XGBoost trained on data with missing-value imputation showed a much lower sensitivity as compared to all others, a typo?

5.     In table 1, could the authors comment on why the diabetic groups did more check-ups throughout a longer follow-up period? Would that introduce bias to the survival analysis?

6.     Line 259 seems to have a reference formatting error. 

Reviewer 2 Report

This is a high quality study conducted on a large populatiom of patients. The manuscript is very Well written. 
The references are up to date, figures and tables are exceptional.

I believe this study will add to the field of diabetes care.

moreover, this field is interesting to a broad readership 

Reviewer 3 Report

The sample of the paper is wide the follow-up is remarkable and the study is well designed. The statistic study is apprppriate. 

I think that the only trouble for this kind of predictive data-base is the great number of parameters and variables to evaluate, that is not simply to respectin real life medicine, but it doesn't affect the good level of the paper

Round 2

Reviewer 1 Report

The authors have addressed all my questions and have substantially improved the manuscript. I have no further comments and I congratulate the authors on this lovely work.